# Global health effects of future atmospheric mercury emissions

Yanxu Zhang [1✉], Zhengcheng Song[1], Shaojian Huang[1], Peng Zhang[1], Yiming Peng[1], Peipei Wu[1], Jing Gu[1], Stephanie Dutkiewicz[2], Huanxin Zhang[3,4], Shiliang Wu[4,5], Feiyue Wang [6], Long Chen [7], Shuxiao Wang [8,9] & Ping Li[10]

Mercury is a potent neurotoxin that poses health risks to the global population. Anthropogenic mercury emissions to the atmosphere are projected to decrease in the future due to enhanced policy efforts such as the Minamata Convention, a legally-binding international treaty entered into force in 2017. Here, we report the development of a comprehensive climate-atmosphere-land-ocean-ecosystem and exposure-risk model framework for mercury and its application to project the health effects of future atmospheric emissions. Our results show that the accumulated health effects associated with mercury exposure during 2010–2050 are $19 (95% confidence interval: 4.7–54) trillion (2020 USD) realized to 2050 (3% discount rate) for the current policy scenario. Our results suggest a substantial increase in global human health cost if emission reduction actions are delayed. This comprehensive modeling approach provides a much-needed tool to help parties to evaluate the effectiveness of Hg emission controls as required by the Minamata Convention.

[1] School of Atmospheric Sciences, Nanjing University, Nanjing, P. R. China. [2] Department of Earth, Atmospheric and Planetary Sciences, Massachusetts Institute of Technology, Cambridge, MA, USA. [3] Department of Chemical and Biochemical Engineering, University of Iowa, Iowa City, IA, USA. [4] Geological and Mining Engineering and Sciences, Michigan Technological University, Houghton, MI, USA. [5] Civil and Environmental Engineering, Michigan Technological University, Houghton, MI, USA. [6] Centre for Earth Observation Science, Department of Environment and Geography, University of Manitoba, Winnipeg, MB, Canada. [7] Key Laboratory of Geographic Information Science (Ministry of Education), School of Geographic Sciences, East China Normal University, Shanghai, P. R. China. [8] School of Environment, State Key Joint Laboratory of Environment Simulation and Pollution Control, Tsinghua University, Beijing, P. R. China. [9] State Environmental Protection Key Laboratory of Sources and Control of Air Pollution Complex, Beijing, P. R. China. [10] State Key Laboratory of Environmental Geochemistry, Institute of Geochemistry, Chinese Academy of Sciences, Guiyang, P. R. China. ✉email: zhangyx@nju.edu.cn

Mercury (Hg) is a global pollutant, and its organic form, methylmercury (MeHg) is associated with neurocognitive deficits in human fetuses and cardiovascular effects in adults[1,2]. Human exposure to MeHg is predominantly via the consumption of food (e.g., seafood and rice)[3,4]. The annual death from the fatal heart attack that is attributable to MeHg exposure is estimated to be over 10,000 in China[5]. Economic losses from intelligence quotient (IQ) decrease of developing brains associated with MeHg exposure has been estimated at $16 billion in the U.S. and European Union[3,5,6]. To protect human health and the environment, the Minamata Convention on Mercury, a legally-binding international treaty, took effect in August 2017 to reduce anthropogenic emissions of Hg (https://www.mercuryconvention.org).

Future projections of global primary anthropogenic Hg emissions vary drastically driven by underlying social-economic and technological change[7,8]. The re-emissions from soils and oceans that receive past atmospheric depositions of Hg (legacy emissions) are also important sources, the magnitude of which is 2-3 times larger than the primary anthropogenic emissions[9,10]. The MeHg exposure is influenced by a chain of processes including atmospheric emission, atmospheric transport and deposition, air-sea exchange, air-land exchange, chemical transformation (especially Hg methylation), food web transfers, and human food intake[11]. These processes are modulated by the fluctuation and change in climate, land-use, ocean circulation, and ecosystem functions[12,13]. Earlier studies do not link emissions to exposure changes[3,14–17]. Later efforts in global Hg exposure modeling have considered only a subset of these processes. For instance, using atmospheric transport models, atmospheric deposition is considered as an indicator for the level of MeHg in seafood[5,11,13]. Zhang et al.[13] included the impact of changing climate, land-use, and land-cover on atmospheric transport and deposition, and Amos et al.[18] and Angot et al.[19] considered the response of land/ocean re-emissions to anthropogenic emission change with a box model.

In this study, we develop a more comprehensive approach to project the change in human MeHg exposure responding to Hg emission changes. We integrate changes in anthropogenic emissions, climate, and biogeochemical cycles. We use a coupled three-dimensional atmosphere/ocean and two-dimensional land model. The Hg/MeHg levels in the environment are used to scale an intake inventory of MeHg for different countries, which are further used to calculate the health impact based on epidemiology-based dose-response relationships (see "Methods" for details). We present a map of MeHg-related health risks for all the countries. Based on this, we translate future Hg emission projections into health risks, and to help parties and stakeholders to evaluate impacts from changes in Hg emissions.

## Results and discussion

**Baseline Hg-related health risk**. We estimate that the global health impacts associated with MeHg exposure for the general population are $117 billion (2020 USD adjusted by purchasing power parity, PPP), contributed by $1.2 \times 10^7$ points of IQ decrements (0.086 point per-fetus) and 29,000 deaths per year at present-day. We include two health endpoints as a consequence of food MeHg exposure: decrement in IQ of newborns and fatal heart attack (FHA) for general populations. The IQ decrement is transferred to lifelong earnings loss based on the projections of the population and economic growth of each country[3,20]. The economic loss from FHA is calculated based on a value of statistical life (VSL) approach, which is scaled by the PPP adjusted per-capita GDP value of individual countries[11]. The exposures from seafood (including fish and aquatic animals), freshwater fish

(also including other aquatic animals), and rice consumption are included here with the MeHg concentrations from literature (see Method for more details and the Supporting Information for detailed data).

We find that the MeHg exposure and health risk are associated with the food intake structures of different countries. Coastal countries with large seafood consumption have the largest MeHg exposure, and rice and freshwater fish consumption are non-negligible in some countries (Fig. S5). The highest per-capita seafood MeHg exposure is found in countries with large seafood consumption, such as the Maldives (33 μg/d), Greenland (16 μg/d), Iceland (15 μg/d), and Kiribati (13 μg/d) (Fig. S5). The national average per-capita seafood consumptions are 190, 89, 74, and 48 kg/y for these four countries, respectively, which are much higher than the global average of 15 kg/y (UN FAO, http://www.fao.org). The lowest risk is found in inland countries with nearly no seafood consumption, such as Ethiopia (0.0018 μg/d), Uganda (0.0093 μg/d), and Chad (0.014 μg/d). The MeHg exposure from rice is the highest in Southeast Asian countries such as Indonesia (1.7 μg/d), Laos (0.90 μg/d), and Cambodia (0.77 μg/d) (Fig. S5). The contribution of rice to MeHg exposure has previously been found in communities relying on rice grown in areas heavily contaminated with Hg[4,21], whereas our findings highlight the potential importance of rice consumption for the general population. We find the contribution from rice could be dominant in inland countries with large rice consumptions, e.g., Nepal (58%), Afghanistan (50%), and Bhutan (45%). The spatial distribution of the consumption of freshwater fish is similar to that of rice, and the exposure is the highest in Asian countries such as Cambodia (6.3 μg/d), Myanmar (3.5 μg/d), and Japan (2.9 μg/d). The MeHg exposure from this pathway is also influenced by the fish MeHg concentrations, which causes relatively high exposure over countries such as Russia (3.5 μg/d) and Finland (3.2 μg/d).

The total health risk of MeHg reflects the total exposure of the above-discussed pathways. The associated IQ decrease per-fetus is the highest in the Maldives (1.2 points), Greenland (0.60 points), and Iceland (0.56 points), where the exposure from seafood is high. The risk is the lowest in Uzbekistan (0.0040 points), Tajikistan (0.0036 points), and Ethiopia (0.0012 points), where the consumption of fish and rice are both low (Fig. 1a). Figure 1c shows the economic loss due to IQ reduction of newborns. Besides MeHg exposure, the loss reflects the birth rate and income level of countries. The US ranks first by losing $12 billion per year, followed by China ($7.3 billion), Japan ($6.2 billion), and Russia ($2.9 billion). The spatial pattern of the deaths associated with FHA (Fig. 1b) is quite different from that of IQ decrement but reflects the total population and baseline FHA incidence. The most deaths are from populous countries with mild to high per-capita risks, such as China (5600 per year), Russia (3200), Indonesia (3200), and India (2300). Taking into the difference in VSL per death puts Russia the first place ($9.1 billion) in economic loss from this pathway, followed by the USA ($9.0 billion), China ($7.7 billion), and Japan ($3.2 billion) (Fig. 1d). Combining the two endpoints results in that the US suffers from a total loss of $21 billion per year, followed by China ($15 billion), Russia ($12 billion), and Japan ($9.3 billion) (Fig. 1e). Asia ($48 billion), Europe ($34 billion), and North America ($23 billion) have 90% of the global health risk, with Africa ($6.4 billion), South America ($5.4 billion), and Oceania ($1.5 billion) contributing the remaining 10%.

Our study estimates the health risk associated with MeHg exposure based on food intake inventory and food MeHg concentrations for individual countries at a global scale. Validation data remain sparse but our estimate generally agrees with previous regional-scale studies for China, US and Europe. Our

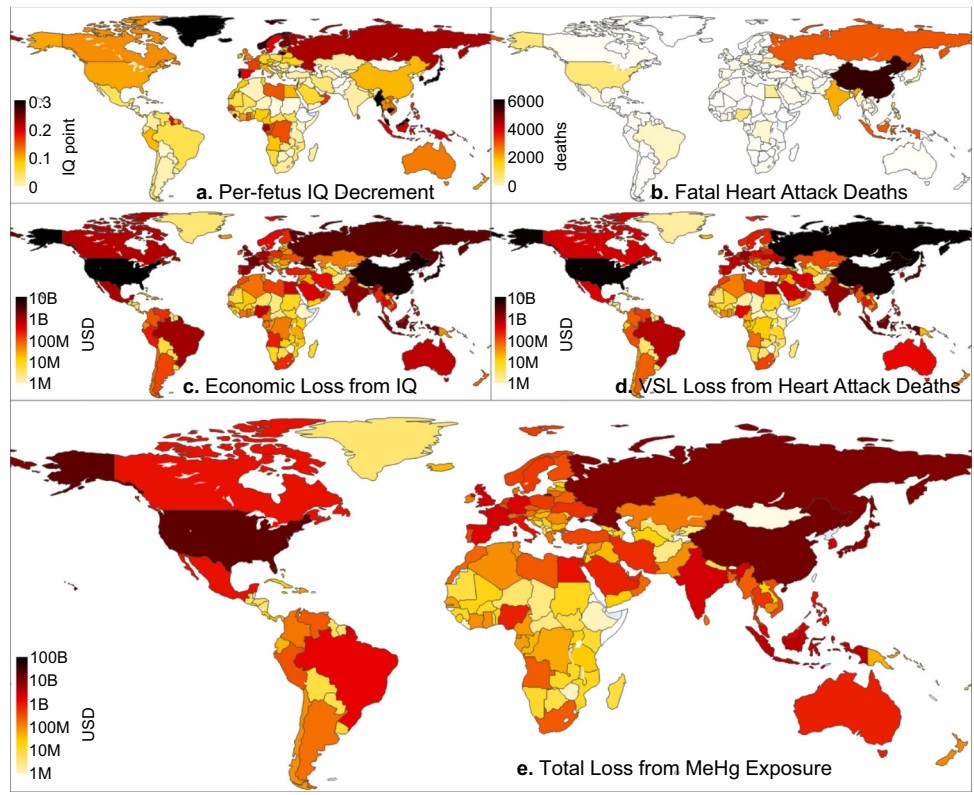

**Fig. 1 Global health impact of methylmercury (MeHg) food exposure at present-day.** (**a**) Per-fetus intelligence quotient (IQ) decrement; (**b**) Fatal heart attack deaths; (**c**) Economic loss from IQ decrease; (**d**) Value of statistical life (VSL) loss from fatal heart attacks; (**e**) Total loss from MeHg exposure (the sum of (**c**) and (**d**)). Economic losses are in United States (US) dollars (2020 value and adjusted by purchasing power parity). The gray area indicates missing data and the color scale of (**c**)–(**e**) is in the logarithmic scale.

results for the Chinese population (0.11 point IQ decrease per fetus and 5600 FHA deaths per year) are slightly lower than Chen et al. (2019) (0.14 points per fetus and 7360 deaths), as we exclude the exposure from other food such as pork, beef, and eggs, which have negligible contribution to the total exposure (except for the eggs and meat from fish-eating seabirds that are consumed by some indigenous populations, e.g., Evers et al. (2003), but there is limited data and may be not important for general populations)[22]. We estimate the per-capita MeHg exposure from seafood consumption for the US population is 11 μg kg body wt$^{-1}$ a$^{-1}$, which agrees with the estimate of Mahaffey et al. (2004) and Sunderland (2007): 7.3–11 μg kg$^{-1}$ a$^{-1}$. We estimate a total of 500,000 points per year of IQ decrements in the US at present-day, which is higher than previous estimates (264,000–285,000 points per year)[6,23], but our estimate (720,000 points) for Europe agrees well with Bellinger et al. (2013), who calculated 640,000 points of IQ loss based on hair Hg concentrations among women of reproductive age.

The modeled spatial pattern agrees with the distribution of Hg biomarkers in general populations from individual countries (Fig. S4). We estimate the hair and blood Hg concentrations based on the total food MeHg exposure and pharmacokinetics models (see Method for details), and evaluate them against available human biomarker data in literature as summarized by Basu et al. (2018). The estimated blood Hg concentrations among the 40 countries where data are available are 2.5 ± 1.8 μg/L (mean ± standard deviation), consistent with measured values (2.2 ± 2.1 μg/L) with a correlation coefficient of 0.71 (Fig. S4). The measured highest mean blood Hg concentrations are found in Greenland (9.2 μg/L), which is well captured by our estimate (10 μg/L, rank = 1st). High blood Hg concentrations are also measured in Cambodia (9.1 μg/L), Spain (6.0 μg/L), Japan (5.1 μg/

L), and South Korea (4.0 μg/L), and our estimates agree with these measurements (4.8, 3.4, 5.5, and 4.6 μg/L, respectively). A lower correlation coefficient (0.53) is calculated for the estimated and measured hair Hg concentrations ($n = 38$), but the estimate (0.40 ± 0.27 μg/g) is within a factor of ~2 from the measured data (0.76 ± 0.48 μg/g) (Fig. S4). In addition to the MeHg exposure, the biomarker level subjects to the variability of pharmacokinetic and intrinsic (such as genetics) factors[24,25]. Overall, our results show that human Hg biomarker levels could be explained by the food Hg exposure for general populations from individual countries, supporting our approach can be used to assess the baseline risk at present-day and its projection in the future.

**Future Hg emissions.** Figure 2 shows global anthropogenic Hg emissions projections under different scenarios. The global total anthropogenic Hg emissions are 1890 Mg yr$^{-1}$ in 2010 with artisanal and small-scale gold mining (ASGM, 37%) and fossil fuel combustion (25%) as the two largest sources followed by non-ferrous metals production (10%) and cement production (9%)[26]. Streets et al. (2009) projected Hg emissions to increase to 4900 and 3900 Mg yr$^{-1}$ in 2050 under the A1B (business as usual) and A2 (a divided world) scenarios, respectively, driven by the increase of coal combustion in developing countries. In a New Policies (NP) scenario, Pacyna et al.[8] projected the emissions to decrease to 1020 Mg y$^{-1}$ in 2035. Part of the emission reduction is from fossil fuel combustion and cement production resulting from the co-benefits of greenhouse gas emission control. The Hg emissions from Hg-containing products are also projected to reduce by 70% in 2035 compared to the 2010 situation, and the use of Hg in ASGM is reduced by 46%[8]. In a Maximum Feasible Reduction (MFR) scenario, the global

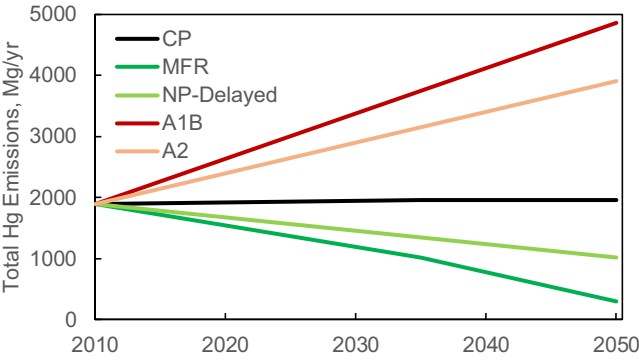

**Fig. 2 Projections of global total anthropogenic emission of mercury during 2010–2050 based on different emission scenarios.** CP (current policy) scenario assumes a near to constant emissions through 2050. A1B and A2 are for business as usual and a divided world scenarios with increasing emissions, respectively. MFR (maximum feasible reduction) assumes the application of the best available technologies and aggressive emission reductions. NP-Delayed (new policy delayed) assumes the 2035 goal of the MFR scenario is delayed to 2050.

emissions are projected to reduce to 300 Mg y$^{-1}$ with the Hg-containing product and ASGM emissions reduced by 95% and 76%, respectively[8]. The best available technologies are assumed for the industrial and energy sectors without constraints in the economy and increasing demand. Although it was considered an unrealistic scenario for 2035[8], we treat it as an optimistic projection for 2050. We refer to this trajectory as the MFR scenario. We also delay the NP scenario from 2035 to 2050 to represent a slower emission reduction pathway (referred to as the NP-Delayed scenario). As a reference, a close to constant emission scenario (except a slight increase of emissions to 1960 Mg yr$^{-1}$ in 2035) is considered following the Current Policy (CP) scenario of Pacyna et al.[8]. This assumes that the increase in emission activity is balanced by the decrease of emission factors due to continuous emission control[8].

**Future MeHg exposure**. We use environmental Hg levels to scale the food Hg concentrations consumed by the general population in the future under various emission change scenarios. The MeHg concentrations in freshwater water fish are influenced by those in their prey and ultimately by river/lake water MeHg levels, which are not explicitly simulated by our integrated model. Instead, we use the atmospheric deposition as a proxy as it is the major source of Hg in surface waters[27]. The rice MeHg concentrations are scaled by soil Hg concentrations due to the strong correlation between them[28]. We use the planktonic MeHg concentrations as a proxy for seafood because the uptake from seawater by plankton represents the largest concentration increase for MeHg bio-magnification in marine food webs[29].

We find that atmospheric Hg deposition and marine planktonic MeHg are highly sensitive to future Hg emissions (Fig. 3). The atmosphere, ocean, and their exchange are simulated by the GEOS-Chem and MITgcm models, both driven by the meteorological/ocean physical data from climate models. The marine plankton biomass and community structure are modeled by an ecosystem model (see "Methods" for more details and a model evaluation against observations is in Figs. S1–3). The model simulates higher deposition over source regions (e.g., East Asia, West Europe, and North America), regions covered with forests that have larger dry deposition velocity (e.g., South America), and over ocean regions with high precipitations (Fig. 3a). Higher planktonic MeHg concentrations are modeled

over productive regions (e.g., high-latitudes and the eastern tropical oceans) (Fig. 3c). The model projects that the MFR and NP-Delayed scenarios reduce the atmospheric deposition in 2050 by 48% and 28%, respectively, compared to the CP scenario. Overall, the decrease of Hg deposition is smaller than that of anthropogenic emissions (85% and 48%, respectively, Fig. 2) because primary emissions only account for 20-30% of total atmospheric Hg emissions[10]. Similarly, the A1B and the A2 scenarios project an increase of primary atmospheric emissions by 150% and 99%, respectively, which only translate to an increase of deposition by 87% and 59%, respectively. The percentage change of planktonic MeHg concentrations is similar to atmospheric deposition, since inorganic Hg, which is the substrate of MeHg in the seawater, is mainly from atmospheric deposition[30]. The percentage changes for different regions are predicted to be fairly uniform. Contrasting to atmospheric deposition, the changes in soil Hg concentrations are much smaller, ranging from −3% to 4% for different scenarios in 2050. This is because of the large mass and long lifetime of Hg in this reservoir[31,32].

**Health effects**. The CP scenario projects a flat trend for the global total IQ decrease until 2050 (an increase from 11.1 to $11.6 \times 10^6$ pts during 2010–2050) (Fig. 4a), reflecting similar trends in both total MeHg exposure (Fig. 3) and new birth number (World Population Prospects: https://population.un.org). We find the changes in future primary anthropogenic emissions are substantially dampened for their health effects. The total IQ decrease in 2050 predicted by the MFR and NP-Delayed scenarios are 24% and 15% lower than that of the CP scenario, respectively, even though the anthropogenic emissions have been projected to decrease by 85% and 48%, respectively. The A1B and A2 scenarios predict a 51% and 34% increase in the IQ effect, respectively, whereas the changes in primary emissions are 150% and 99%, respectively.

The global population is projected to increase by ~40% to 9.7 billion in 2050 (World Population Prospects), which translates to the projected FHA deaths associated with MeHg exposure by 43% to 40,000 per year for the CP scenario. This results in a cumulative death of 1.6 million during 2010-2050. The increase in total population also cancels the decreasing trend in per-capita exposure of the MFR and NP-Delayed scenarios. The projected trajectory for the deaths of these two scenarios is quite flat, with a cumulative death of 1.4 and 1.5 million, respectively (Fig. 4b). In contrast, the projected deaths for A1B and A2 scenarios are 120% and 94% higher than the level in 2010, amounting to a cumulative death of 2.0 and 1.9 million, respectively.

The economic valuation of these two health endpoints relies on the projection of the global economy. We adopt the middle-of-the-road pathway projected by the Shared Socioeconomic Pathways (SSP2) in the 21st century (https://tntcat.iiasa.ac.at/SspDb). Due to the increase in per-capita GDP, the total economic loss of the CP scenario is projected to increase by a factor of 2.3, and the loss for the strictest emission reduction scenario, MFR, also increases by a factor of 1.4 (Fig. 4c). The cumulative economic loss for the CP scenario is $19 trillion (2020 USD, discounted to 2050 at a rate of 3%). The projected health benefits of the MFR and NP-Delayed scenarios compared to the CP scenario are $2.4 trillion and $1.5 trillion, respectively. On the other hand, the A1B and A2 scenarios will result in an additional loss of $4.9 trillion and $3.3 trillion, respectively. The two health endpoints contribute roughly equally to the total loss; however, the contribution from the VSL of FHA becomes more dominant (60%) in 2050 due to a faster increase in total population than new birth.

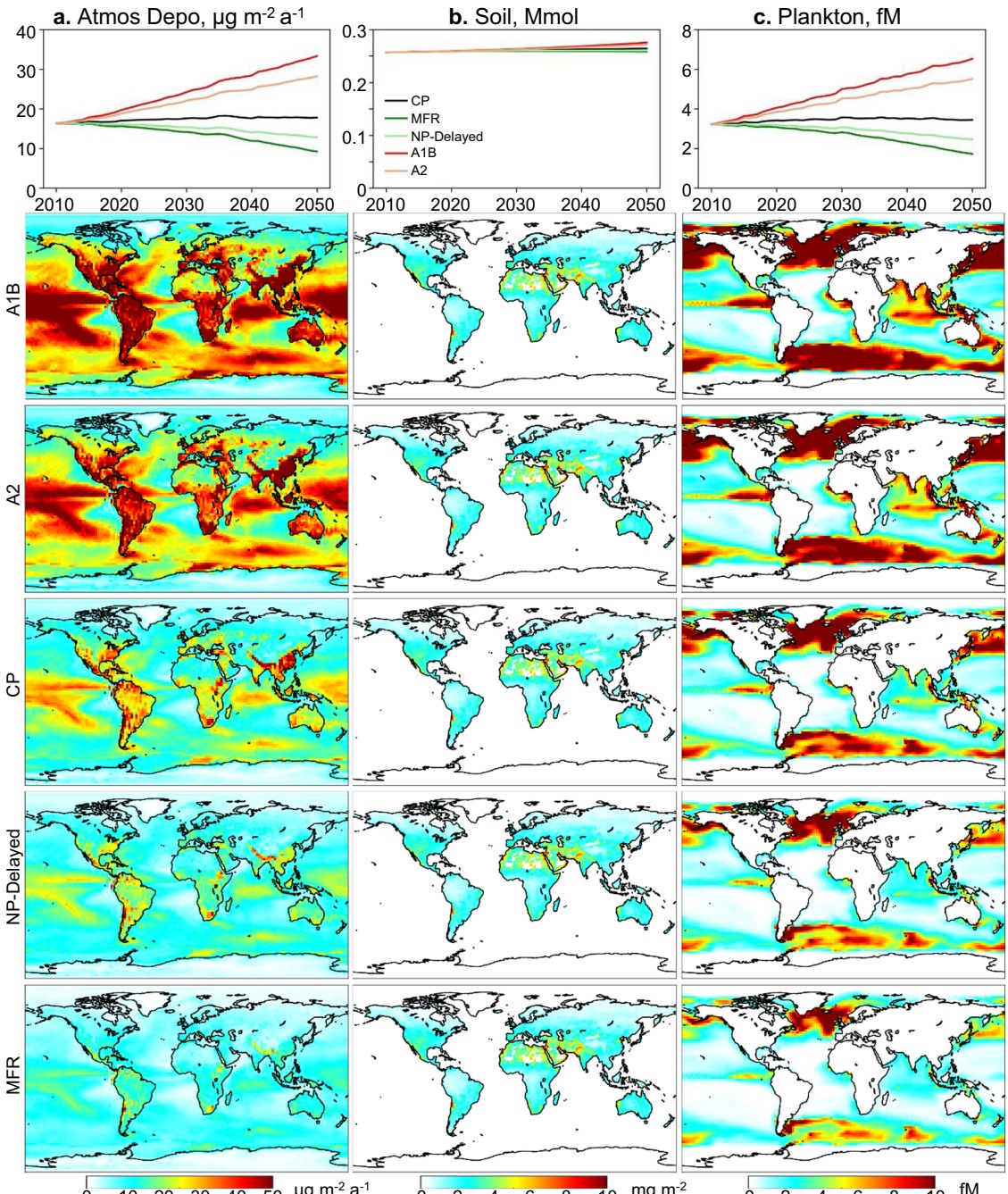

**Fig. 3 Projected mercury levels in the environment in 2050.** The column (**a**), (**b**), and (**c**) is for atmospheric deposition, soil, and marine plankton, respectively. For each column, the top panel shows the trend of global mean values, while the lower panels are the spatial distribution for the five scenarios (with emission from highest to lowest): A1B (business as usual), A2 (divided world scenario), CP (current policy), NP-Delayed (new policy delayed), and MFR (maximum feasible reduction).

**Uncertainty**. We assess the uncertainty and variability of the health effects projected by our integrated model by identifying key driving factors, including food consumption data, food MeHg concentrations, dose-response parameters linking MeHg exposure and health effects, and economic valuation (Fig. 5). We rely on the database of the United Nations' Food and Agriculture Organization (FAO, http://www.fao.org) for food consumption. Compared with national data, the two data sources generally agree within a factor of 2 (Figure S6). This reflects both the different survey methods and variability among the population[33,34]. This results in a variability of cumulative

economic loss for the CP scenario as $10 to $27 trillion (95% confidence interval in 2020 value and realized in 2050, same thereafter). This variability also propagates to the estimated benefits (or extra costs) for other scenarios (Fig. 5). By considering the log-normal distributions of food MeHg data, the cumulative effects for the CP scenario would range from $12 to $31 trillion. This indicates that the food intake and MeHg data contribute roughly equally to the uncertainty of exposure calculation. We find that the dose-response functions between MeHg intake and health effects have the largest contribution to the uncertainty, ranging from nearly $7.8 to $47 trillion. This reflects

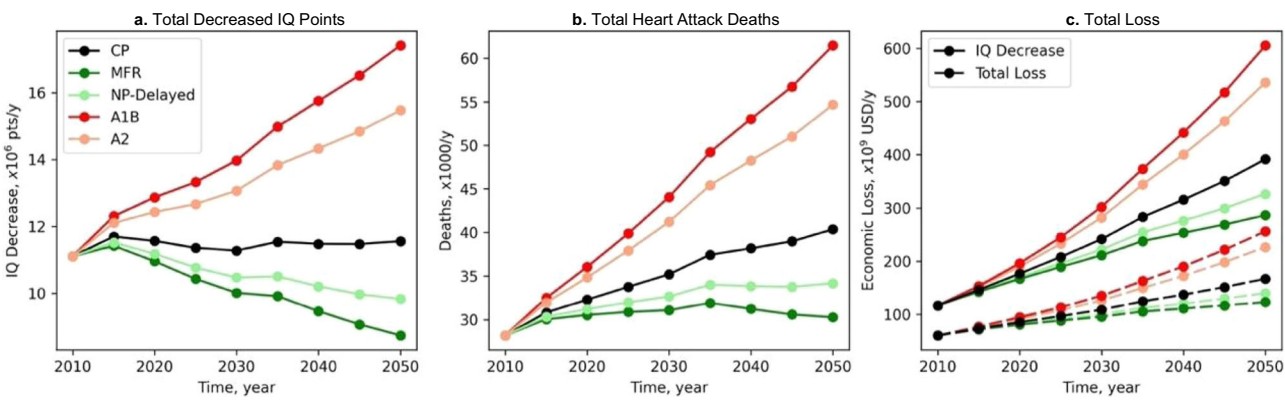

**Fig. 4 Trajectories of global annual health effects associated with different future emission scenarios.** (**a**) Total intelligence quotient (IQ) decrements of newborns; (**b**) Total heart attack deaths; (**c**) Economic valuation of health effects: total valuation (solid lines) and from IQ decrements (dashed lines). Five scenarios are included: A1B (business as usual), A2 (divided world scenario), CP (current policy), NP-Delayed (new policy delayed), and MFR (maximum feasible reduction).

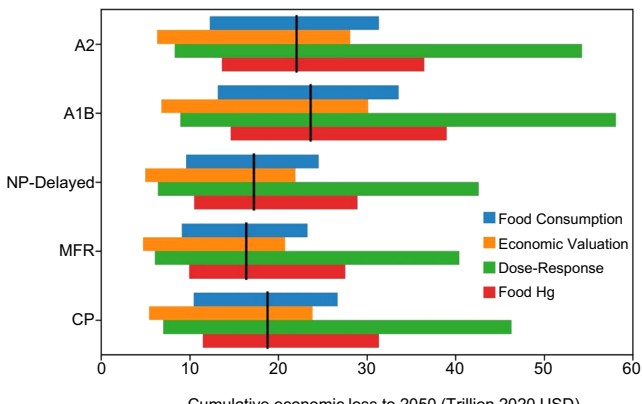

**Fig. 5 Range in cumulative health impacts (unit: US dollars in 2020 value) to 2050 for the CP (current policy), MFR (maximum feasible reduction), NP-Delayed (new policy delayed), A1B (business as usual), and A2 (divided world scenario) scenarios.** Bars indicate the sensitivity of cumulative health effects to high and low case assumptions for uncertain parameters (as 95% confidence intervals): food consumption, economic valuation, dose-response parameterization, and food methylmercury concentrations. The black lines are our best estimates.

the large variability in the coefficients for IQ decrement and heart attack risk per unit hair Hg increase[11,35], despite convincing evidence for the association between MeHg exposure and human health impact[36]. The pharmacokinetics parameters to link food exposure to blood and hair Hg levels play a much smaller role with a variability of 10–20%. Another source of uncertainty comes from the parameters for economic valuation, especially the VSL of heart attack deaths (a factor of 10)[11]. Using high and low assumptions for the economic valuation leads to a range of $5.8–24 trillion for the health effect of the CP scenario. By taking a Monte Carlo approach (see Methods), we also calculate the overall uncertainty range as $4.7–54 trillion.

Our ability to model the MeHg exposure and risk is limited by existing scientific knowledge and data, such as the food web dynamics of MeHg in higher trophic levels and the dose-response relationships between MeHg exposure and its health effects (Fig. 5). The future Hg emissions to water and soil are subjected to change[37]. We only consider the general populations, but not the so called high exposure population groups[24]. The fishery harvest and human food consumption patterns will also change in the future[38]. Our results do not show strong interannual variability for environmental Hg levels on a global scale, but the

change in dietary structure and food web dynamics in high trophic levels that are not covered in this model may amplify these variabilities, especially at regional scales[39]. The permafrost stores a large amount of Hg and may serve as a potential Hg source as a consequence of thawing[31]. There are likely other health endpoints not considered in this study due to the limited epidemiological data[36]. Our assessment is thus considered illustrative and not a comprehensive projection of impacts. However, much uncertainty of the model framework could be reduced using a similar methodology as science and data evolve.

**Policy implications.** This study develops and applies a comprehensive climate-atmosphere-land-ocean-ecosystem and exposure-risk model framework for global toxic pollution from Hg. We show that the annual global health risk associated with MeHg exposure at present-day is $117 billion (2020 value), contributed by $1.2 \times 10^7$ points of IQ loss and 29,000 heart attack deaths per year. By 2050, the cumulative health effects are projected to be $19 ($10–27 as uncertainty range) trillion (discount rate of 3% to 2050). Compared to the CP scenario, the MFR and NP-Delayed scenarios have benefits of $2.4 and $1.5 trillion, respectively, while the A1B and A2 scenarios have additional losses of $4.9 and $3.3 trillion, respectively.

Food intake structure is an important factor for MeHg exposure and risk. Globally, seafood consumption contributes 56% to the total MeHg exposure, with freshwater fish and rice contributing 34% and 10%, respectively. Coastal and island countries with access to more seafood have the largest seafood consumption and they will have the greatest health benefits if Hg emissions are reduced in the future. Freshwater fish consumption is the highest in Asian countries, where fish is often raised in rice paddies[40]. The rice consumption in these countries is also high. Despite the elevated MeHg exposure risk, the overall health effects of fish consumption may be positive if considering the intake of n-3 polyunsaturated fatty acids, vitamins, and other nutrients[41]. Another important influencing factor is the trophic level of fish/aquatic animals. The mean Hg levels vary for ~10 times between the lowest and highest trophic levels, much larger than the impact of water types and whether wild-caught or farm-raised (Fig. S7). Dietary guidance on fish selection but not the total fish consumption is the rule of thumb to minimize the overall health risks, especially considering the nutrient effects of fish[42,43]. For countries with the least MeHg exposure as found in this study, such as Ethiopia, Tajikistan, and Afghanistan, which are listed as the countries with serious levels of hunger (Global Hunger Index: https://www.globalhungerindex.org). In these

countries, the MeHg exposure risk of fish consumption is even more outweighed by its nutrient benefits[44]. We suggest that the Hg level in rice is the most recalcitrant to emission reduction among the three major food categories, and the global contribution from rice consumption could increase to 23% in 2050 under the MFR scenario, which makes limiting rice consumption may be a more important Hg exposure mitigation strategy then.

This study focuses on the MeHg exposure of the general population. Significantly higher exposure and biomarker levels are found for populations exposed to Hg from point sources (e.g., ASGM workers) and populations with high seafood consumption (e.g., Arctic populations that consumes a lot of marine mammals, tropical riverine communities, and coastal and/or small-island communities)[24]. Although we do not include these two groups due to the lack of global-scale data, to include them will supplement our estimate of MeHg exposure and health risk. With the future improvement of spatial resolution in exposure and risk modeling, our model could also be a useful tool to identify populations that are vulnerable to Hg exposure[45].

We show that the cumulative health effects realized within the 2050 time horizon are not responding linearly with emission changes. The MFR scenario with a rather low emission level in 2050 (300 Mg yr$^{-1}$) is only 13% lower than the CP scenario. This is associated with the relatively small change in emissions between the two scenarios in the early years (e.g., during 2010–2025), which contributes a large portion to the cumulative health effect realized in 2050 due to its compound interest. Delaying the MFR scenario (i.e., the NP-Delayed scenario) would further reduce the benefit by 38%, not to mention the substantially increased health effects projected for the A1B and A2 scenarios. Even though these estimates are very sensitive to choices of the temporal scope of analysis and evaluation parameters (e.g., VSL, discounting rate)[11], our results demonstrate the necessity of emission reduction sooner.

The inclusion of land and ocean in our model enables us to directly model the contribution of legacy sources (re-emissions from soil and ocean) without using a scale function[46] or box models[19]. The inclusion of the ocean model coupled with the plankton ecosystem in our integrated model is of great interest because seafood consumption is the major exposure pathway in most countries (Fig. S5). The marine plankton and soil Hg concentrations are better proxy data to scale the future change of seafood and rice Hg levels, respectively, than atmospheric deposition that is often employed in previous studies[5,11].

Our model framework provides a much-needed tool for parties to evaluate the effectiveness of the implementation of the Minamata Convention, especially to assess the response of environmental Hg levels to emission reduction and its implications to human exposure and health risk. Detailed scenario studies using our model framework could be conducted to evaluate and prioritize the health benefits of individual policy measures. For example, as the largest emission source, the control of ASGM is left decided for individual parties and the future emissions have large uncertainty. Our model framework would assist related countries to make their national action plans.

## Methods

**Mercury transport model**. We develop a model framework to simulate the fate and transport of Hg in the Earth system that includes climate, atmosphere, land, ocean, and marine ecosystem (Fig. 6). Three-dimensional atmospheric (GEOS-Chem) and oceanic (MITgcm) transport models for Hg are coupled online with a two-dimensional terrestrial mercury model (GTMM). These models are driven by predicted meteorological and ocean circulation data from climate models (GISS GCM ModelE2 and IGSM, respectively). Biogeochemical parameters important for Hg transformation are taken from a marine plankton ecosystem model (Darwin), which is also driven by the IGSM model. The details of these models are elaborated below.

We use the output of the Integrated Global System Modeling (IGSM) framework for the future climate simulated by Sokolov et al.[47] and Dutkiewicz et al.[48]. Briefly, the model framework includes a three-dimensional ocean model that has a horizontal resolution of 2° × 2.5° and 22 vertical levels from 10 m in the surface to 500 m at depth, and a two-dimensional (latitude and height) atmosphere physical and chemical model. The framework has a terrestrial component with hydrology, vegetation, and natural emissions. The model is run with a pre-industrial level of greenhouse gas concentration for 2000 years as spin up and then for 1860–2000 with observed GHG levels. For the 21st century, a business as usual scenario (close to IPCC AR5 RCP8.5 scenario) is assumed for anthropogenic emissions. We use the IGSM archived monthly mean ocean physics data such as seawater temperature, ocean current velocities, and mixing conditions to drive the MITgcm model[48].

As the IGSM only contains a two-dimensional atmosphere module, we use the archived future climate data simulated by the NASA Goddard Institute for Space Studies (GISS) general circulation model (GCM) (ModelE2) to drive the GEOS-Chem model[49]. The model has a horizontal resolution of 2° × 2.5° for the atmosphere, land surface, ocean, and sea ice models. The three-dimensional atmosphere model has 40 vertical levels from the surface to 0.002 hPa (~85 km altitude). The greenhouse gas concentrations are specified following the IPCC AR5 RCP8.5 scenario. The meteorology fields such as temperature and precipitation are archived with a frequency of 3–6 h. The discrepancies between these two climate models are minimal due to the similar pathways of greenhouse gas concentrations in the 21st century.

The ocean biogeochemistry and ecosystem data are from the Darwin model within the MITgcm framework simulated by Dutkiewicz et al.[48] during the 21st century. This model is driven by the archived ocean physics fields from the IGSM. The transport of inorganic and organic forms of carbon, nitrogen, phosphorus, iron, and silica are included. The model includes six phytoplankton functional groups and two zooplankton grazers (namely diatoms, other large phytoplankton, diazotroph, coccolithophore, *Prochlorococcus*, *Synechococcus*, and small and large herbivorous plankton). The model simulates biogeochemical processes including phytoplankton growth, zooplankton grazing, zooplankton mortality, and the formation and transformation of particulate and dissolved organic matters. The monthly mean concentrations of organic carbon and plankton biomass, and the rates of plankton growth, grazing, and mortality are archived to drive the Hg component of the model.

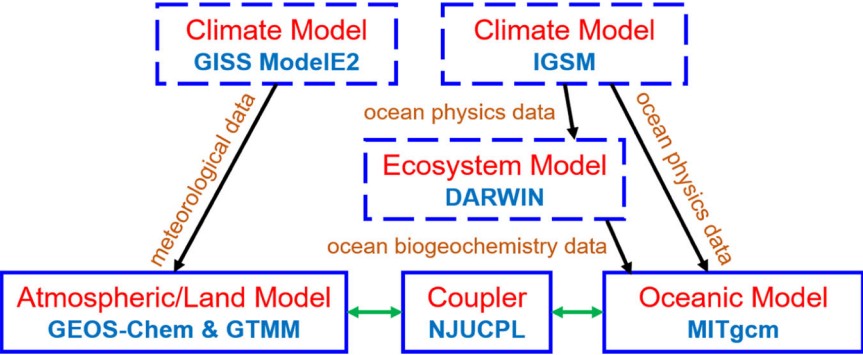

**Fig. 6 The model framework used in this study.** Blue boxes represent models while arrows indicate data flows. Models with dashed boxes are run by previous studies and we directly use their archived output, while those with solid lines are run in this study. Black arrows mean feeding of archived data from one model to another (i.e., models are run separately), while green arrows mean running two models simultaneously with online data exchange.

We simulate the chemistry, transport, and trophic transfer of Hg in the ocean using the MITgcm following Zhang et al.[30]. The model is driven by ocean physics data from the IGSM and biogeochemical parameters from the Darwin model. The model has the same grid as the ocean component of the IGSM. The model simulates 14 tracers that include elemental Hg, oxidized Hg, monomethylmercury (MMHg), dimethylmercury (DMHg), particulate-bound inorganic Hg, and particulate-bound MMHg, and MMHg in plankton (six phytoplankton groups and two zooplankton types). The model includes a detailed photo- and biological mediated redox chemistry of inorganic Hg, and the transformation with methylated Hg species. The bioaccumulation and biomagnification of MMHg in the marine plankton food web is simulated following Zhang et al.[30].

The atmospheric chemistry and transport of Hg are simulated by the GEOS-Chem model following Horowitz et al.[10]. The model is based on version v9-02 (www.geos-chem.org). The model has the same resolution as the GISS GCM ModelE2 model and is driven by its archived meteorology fields. The simulated chemistry includes three forms of Hg in the atmosphere: gaseous elemental, gaseous oxidized, and particle-bound Hg. The redox chemistry of Hg is modeled following Horowitz et al.[10]. The chemistry includes two-stage oxidation of elemental Hg by bromine atoms and photoreduction of oxidized Hg in cloud droplets. Concentrations of related chemical species are taken from GEOS-Chem simulation of tropospheric oxidant-aerosol chemistry[50]. The model also includes geogenic and biomass burning sources, as well as reemissions from snow reservoirs[10].

The soil pool and land-atmosphere exchange of Hg are simulated using the GTMM model[32]. The model has a horizontal resolution of 1°x1° and covers the top 30 cm of soils. The model takes the atmospheric deposition of both elemental and oxidized Hg from GEOS-Chem as input. This part of Hg is assumed to be loosely adsorbed to soil and leaf surfaces and undergoes photoreduction and revolatilization. Part of the Hg can reach to soil pool through litterfall. The model considers four soil Hg pools that are tied to carbon pools with characteristic turnover time ranged from $10^0$ to $10^4$ years. Hg bound to different soil carbon pools can be transformed between each other, and Hg is released to the atmosphere when the soil carbon is respired by microbial activities. The model is run for 30,000 years with atmospheric deposition at pre-industrial levels before ramped up with monthly deposition from 1840 to 2000[32].

The GEOS-Chem, GTMM, and MITgcm Hg models are online two-way coupled using a coupler (NJUCPL) following Zhang et al.[51]. With a frequency of 60 min, atmospheric Hg concentration and deposition data are passed from GEOS-Chem to GTMM and MITgcm, and soil re-emission and ocean evasion fluxes are passed from GTMM and MITgcm to GEOS-Chem, respectively. The initial conditions of these models are taken from previous simulations for the present-day[10,30,32]. The total anthropogenic Hg emissions for the future are taken from Streets et al.[7] and Pacyna et al.[8]. Five policy scenarios are developed for the emissions during 2010–2050 (Fig. 2). These global total emissions are spatially distributed to each model grid based on the WHET emission inventory for 2010[52]. The spatial distribution and speciation of Hg emissions are assumed to keep constant during the model period. The model is run from 2010 to 2050 for each emission scenario.

**Mercury exposure modeling.** Figure 7 summarizes the approach we use to estimate MeHg exposure and risk. The MeHg exposure via three food categories is considered in this study: seafood, freshwater fish (including aquatic animals), and rice. Other types of food are ignored because of less data and much lower MeHg concentrations. The per-capita consumption of different food categories (including rice, total fish and aquatic animals) for each country is taken from the database of the Food and Agriculture Organization of the United Nations (UN FAO, http://www.fao.org).

A database for the average MeHg concentrations of these food categories is developed by collecting available data from the literature (a full list of literature is provided in the Supporting Information). There is a total of 210,000 data points (208,000 for fish/aquatic animals and 6,400 for rice) collected from 395 publications (data handled by Microsoft Excel 2019). We exclude the data points near point sources or contaminated sites. The fish/aquatic animals are further divided into two categories: farm-raised and wild-caught with fractions from the UN FAO database. Due to the large concentration variability and the lack of fish/aquatic animals consumption data for individual species, we group the fish/aquatic animals into four tropic level bins: 2–2.5, 2.5–3.5, 3.5–4.5, and 4.5–5, and the geometric mean of MeHg concentrations for each trophic level bin is calculated. The trophic level of each fish with reported MeHg concentration and consumption data is from the Fishbase Database (https://www.fishbase.org). The fraction of fish consumption for each trophic level bin is estimated based on the marine trophic index if detailed consumption inventory is missing (http://www.seaaroundus.org/mti-fib-rmti/). We use the global geometric mean concentrations for countries without data. The total MeHg exposure ($E$) for the general population from each country is calculated as:

$$E = \sum_{i=1}^{4} I_{i,j}^{\text{FW fish}} C_{i,j}^{\text{FW fish}} + \sum_{i=1}^{4} I_{i,j}^{\text{seafood}} C_{i,j}^{\text{seafood}} + I^{\text{rice}} C^{\text{rice}} \quad (1)$$

where $I$ and $C$ are for food intake and MeHg concentrations, respectively, for each category [freshwater (FW) fish, seafood, and rice] and trophic level bin $i$ (2–2.5, 2.5–3.5, 3.5–4.5, and 4.5–5) (Fig. 7). The agreement with human biomarker data suggests that our simplified exposure model works reasonably well.

**Future mercury exposure.** We scale the future population exposure of MeHg based on the exposure level at present-day and the model-projected environmental Hg levels (Fig. 7). Due to the lack of data, the food consumption pattern is held constant during 2010–2050. The freshwater fish and rice MeHg concentrations for individual countries in a given year are assumed to be proportional to the average total Hg atmospheric deposition ($D$)[5,11] and total soil Hg concentration ($S$)[28] in the corresponding country, respectively. For the seafood, we assume the MeHg exposure of each country for a given year is proportional to the global average plankton MeHg concentrations weighted by the spatial distribution of fish harvest ($P$)[53]:

$$C_{\text{year}}^{\text{FW fish}} = C_{2010}^{\text{FW fish}} \frac{D_{\text{year}}}{D_{2010}} \quad (2)$$

$$C_{\text{year}}^{\text{rice}} = C_{2010}^{\text{rice}} \frac{S_{\text{year}}}{S_{2010}} \quad (3)$$

$$C_{\text{year}}^{\text{seafood}} = C_{2010}^{\text{seafood}} \frac{P_{\text{year}}}{P_{2010}} \quad (4)$$

**Human health impact.** We include two health endpoints in benefit estimates: decrement in IQ of newborns and fatal heart attack (FHA)[5,11,16]. A linear dose-response relationship without thresholds is recommended by the National Research Council (NRC) between MeHg intake and fetal IQ decrements[1]:

$$\Delta IQ = \gamma \lambda \beta \times \Delta EDI \times BW \quad (5)$$

where $\Delta IQ$ is the changes in IQ (points), $\Delta EDI$ is the changes in estimated daily intake (EDI) of MeHg, and BW is the average body weight for female adults. The coefficients $\beta$ (0.6 µg L$^{-1}$ per µg day$^{-1}$), $\lambda$ (0.2 µg g$^{-1}$ per µg L$^{-1}$), and $\gamma$ (0.3 IQ points per µg g$^{-1}$) convert from MeHg intake to blood concentration, blood concentrations to hair concentrations, and hair concentrations to IQ decrements,

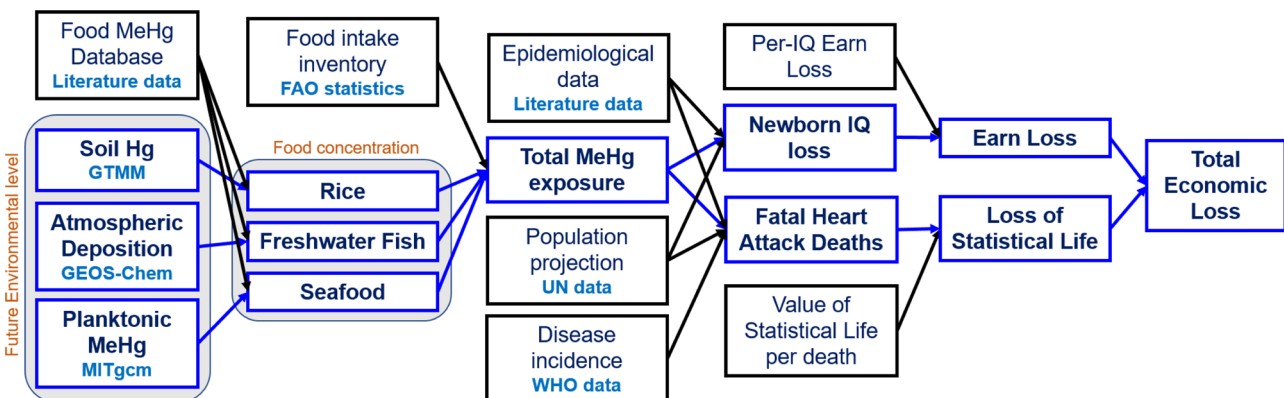

**Fig. 7 Diagram for the approach and data sources for methylmercury exposure and risk calculation in this study.** Blue frames represent model calculated variables, while black ones are for literature data.

respectively. The total IQ decrements are calculated by multiplying per-fetus IQ decrements and the number of newborns, which are taken from the World Population Prospects by the UN database (https://population.un.org). This database includes projections of the total population of both sexes and newborns for each country to 2100.

A log-linear dose-response relationship between MeHg intake and Hg-related fatal heart attacks is used in this study[5]:

$$\Delta \text{CF} = \sum_g \text{POP}_g \times \text{Cf}_g \times \omega \times \left(1 - e^{-\varphi \lambda \beta \times \Delta \text{EDI} \times \text{BW}}\right) \quad (6)$$

where $\Delta \text{CF}$ is the changes in the deaths from fatal heart attacks associated with MeHg exposure, $\text{POP}_g$ is the population of gender $g$ (male and female) from World Population Prospects, and $\text{Cf}_g$ is the age-adjusted incidence of FHA of gender $g$ from Global Health Estimates by World Health Organization (http://www.who.int/healthinfo/global_burden_disease). The coefficient $\varphi$ (0.066 per µg g$^{-1}$) converts hair MeHg concentrations to fatal heart attack risks. The subjective coefficient $\omega$ (0.33) represents the probability of the causality of the associations, reflecting the substantial uncertainties due to limited epidemiological studies.

For model evaluation, we also calculated the averaged Hg concentrations in the blood ($C^{\text{blood}}$) and the hair ($C^{\text{hair}}$):

$$C^{\text{blood}} = \beta \times \text{EDI} \times \text{BW} \quad (7)$$

$$C^{\text{hair}} = \lambda \times C^{\text{blood}} \quad (8)$$

The modeled Hg biomarker concentrations for individual countries are compared with the geometric mean of measured data (961,000 data points) from 245 publications for 83 countries and regions (a full list of literature is provided in the Supporting Information). The blood and hair Hg concentrations for general population are used by excluding high exposure group data (high fish consumption population or population exposed to point Hg sources such as ASGM and Hg mines).

**Economic valuation.** The IQ decrements are converted to monetary values using $18,832 (2008 value) per IQ point normalized by the ratio between the PPP-adjusted GDP per capita in each country and the US[3]. The economic loss from FHA deaths associated with MeHg exposure is calculated by a value of statistical life (VSL) approach. We adopt a VSL per death of $6.3 million (2005 value) following Giang and Selin[11]. This value is also normalized by the PPP-adjusted GDP per capita in each country. The sum of these two endpoints is calculated as the total economic loss. The economic data is taken from the shared socioeconomic pathways database that projects the GDP growth for each country in the 21st century (https://tntcat.iiasa.ac.at/SspDb). We use the SSP2 scenario that assumes a median level of GDP growth rate. A discount rate of 3% is used to realize the economic loss from 2010-2050 to 2050[11].

**Uncertainty analysis.** We consider the contribution of the data and parameters for food consumption, food MeHg concentrations, dose-effect relationship, and economic valuation to the total uncertainty. We compare the food consumption data from UN FAO with national datasets (details available in the Supporting Information). The difference between them is used to represent the uncertainty range of the FAO datasets. For food MeHg concentration, we use the variability of the log-transformed concentrations in each food category to represent its uncertainty. We use the ranges (or standard deviations) of the dose-effect relationship between MeHg exposure and its health effect summarized by Chen et al.[5] and Giang and Selin[11]. For per-IQ earn loss, we use a high- and low-end value of $18,832 and $8013, respectively[11]. The VSL per death ranges from $1 to $10 million following Giang & Selin[11]. The overall uncertainty is estimated by a Monte Carlo approach. The health risk calculation is repeated for 1000 times with randomly sampled parameters for these four factors following Chen et al.[5]. The 2.5% and 97.5% percentiles of the calculated risk are taken as the overall uncertainty range (i.e., 95% confidence interval). The exposure and risk calculation and the associated uncertainty analysis are conducted using Python 3.8.

**Reporting summary.** Further information on research design is available in the Nature Research Reporting Summary linked to this article.

## Data availability

All data generated or analyzed during this study are available in the Supplementary Information and the research group website: https://www.ebmg.online/mercury. FAO/WHO global individual food consumption database: http://www.fao.org/nutrition/assessment/food-consumption-database/en/. World population prospects: https://population.un.org. Shared socioeconomic pathways database: https://tntcat.iiasa.ac.at/SspDb. Global hunger index: https://www.globalhungerindex.org. Fishbase database: https://www.fishbase.org. Marine trophic index: http://www.seaaroundus.org/mti-fib-rmti/. Global health estimates: http://www.who.int/healthinfo/global_burden_disease.

## Code availability

All model code is available at the research group website: https://www.ebmg.online/mercury.

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

## Acknowledgements

We gratefully acknowledge financial support from the National Natural Science Foundation of China (NNSFC) 41875148, the Chinese Academy of Science Interdisciplinary Innovation Team (JCTD-2020-20), Jiangsu Innovative and Entrepreneurial Talents Plan, the Collaborative Innovation Center of Climate Change, Jiangsu Province. We thank Aijun Ding, Bin Wang, Guoxing Li, Qingru Wu, Lars-Eric Heimbürger, Noelle Selin, and Elsie Sunderland for helpful discussions. We are grateful to the High Performance Computing Center (HPCC) of Nanjing University for doing the numerical calculations in this paper on its blade cluster system.

## Author contributions

Y.Z. designed and conducted this study. Y.Z., Z.S., S.H., P.Z., Y.P., P.W., and J.G. compiled literature data. S.D, H.Z., S.W., L.C., and P.L. provided model tools or datasets. Y.Z. led the manuscript writing with supports from S.D., H.Z., S.W., F.W., L.C., S.W., and P.L.

## Competing interests

The authors declare no competing interests.
