## [Peer Review File · Nature Communications]

REVIEWER COMMENTS

Reviewer #1 (Remarks to the Author):

The submission by Zhang et al. is an important contribution which will be of interest to a global audience. The article effectively communicates the magnitude of the health and economic burdens created by worldwide mercury emissions. The article is important because it is a rare example of a quantitative model that integrates fundamental processes from emissions to transport to exposure to health impact to economic damages. This submission is likely to be highly influential from a decision/policy support perspective.

My biggest concern is that the authors sometimes describe fish consumption only in terms of its risk profile from MeHg without considering that, for most people, fish consumption is overall a good thing (even considering MeHg exposures). For example, the authors seem to suggest that only in countries facing hunger crises do the benefits from fish outweigh the risks. I am not sure if this is what the authors meant to say but I strongly disagree with that message. I provide several suggestions below about how this message can be reframed/fine-tuned to capture more of the nuances around benefits from fish in view of risks from MeHg. I believe all of those comments can be addressed with no impact on the quantitative analysis.

I have also included several other miscellaneous technical comments/questions (again, mostly on the exposure/health side) that could be opportunities for the authors to clarify their methods or possibly adjust their analysis slightly.

Overall, I highly encourage you to accept the submission by Zhang et al. subject to the revisions noted below.

1. Abstract

“the most optimistic scenario (maximum feasible reduction, MFR) leads to Hg levels in the freshwater and marine biota half of the present-day levels.”

- By when? By 2050?

2. Introduction

“Mercury (Hg) is a global pollutant that is associated with impaired neurocognitive deficits in human fetuses and cardiovascular effects in adults (Axelrad et al., 2007; Roman et al., 2011).”

- You mean neurocognitive deficits/impacts or impaired neurodevelopment. An impaired deficit would be a good thing!
- I assume you are referring here to impacts of relatively low levels of methylmercury specifically. Higher MeHg exposures can cause neurological impacts even among adults. Other species of Hg (e.g., inhalation of Hg₀ vapor) also has various impacts, including neurological impacts.
- It is important at this stage to know whether you are focusing specifically on MeHg or whether you are talking more broadly about all Hg exposures.

“Human exposure to Hg is predominantly via the consumption of food (e.g., seafood and rice) that contain methylmercury (MeHg), the most toxic form of Hg (Bellanger et al., 2013).”

- Unless I am mistaken, the Bellanger reference does not say anything about non-MeHg exposures to Hg. Therefore, it can't be used to support a statement that most Hg exposures are from MeHg. As far as I read, it only says that most MeHg exposures are from seafood.

“The annual death from the fatal heart attack that is attributable to MeHg exposure is estimated to be over 10,000 in China and the U.S. (Chen et al., 2019; Giang & Selin, 2016).”

- Can you double check this reference? As far as I read, Giang and Selin do not provide an estimate for the baseline fatalities from MeHg exposure.

3. Results and discussion

“lifelong earn loss”

- “earn” should be “earnings”

“freshwater fish consumption are not negligible in some countries”

- *Non*-negligible?

“The highest seafood MeHg exposure is found in countries with large seafood consumption, such as the Maldives (33 µg/d),...”

- Is this µg/day per capita?

“We find that the population's dietary choices are vital factors influencing MeHg exposure and health risk.”

- This paragraph is slightly problematic because it presents a one-dimensional view of fish intake as a vector for MeHg risk. Yes, people who eat the most fish will generally have the highest MeHg exposures, but it does not necessarily follow that they will have the greatest overall risks; depending on the fish they consume, it is likely that, overall, intake of n-3 fatty acids, vitamin D and other nutrients will exert net health protective effects considering relevant alternative sources of protein. (For example, see Calder et al. 2019 where epidemiological modeling demonstrated that even in settings of elevated MeHg, reducing fish consumption almost certainly increases overall risks: <https://www.ncbi.nlm.nih.gov/pmc/articles/PMC6317887/>). A better framing for this paragraph would be to observe that individuals with the highest fish consumptions and highest MeHg intakes would have the most to gain from reduced Hg emissions and MeHg exposures because they would benefit from greater net benefits of fish intake (risks would go down while benefits would stay the same for a given intake of fish).

“we exclude the exposure from other food such as pork, beef, and eggs, which have limited MeHg concentration data”

- You should say that you exclude these foods not because they have limited data but because MeHg is negligible compared to fish and seafood. (This is the reason there is limited data – they are not important contributors to overall MeHg exposures.)

The exception here could be eggs and meat from seabirds like loons and other birds that eat fish. In some populations (e.g., Indigenous people in North America) these can be important contributors. I agree that for these foods, there might be real exposures but there is limited data. I know there is lots of loon Hg/MeHg data available for northeastern North America (mostly by Dave Evers, e.g., <https://link.springer.com/article/10.1023/A:1022593030009>) but data for other seabirds has been hard to find in my experience.

It may be good to make a distinction between 1) animals with no link to aquatic food webs (e.g., pork, beef, most eggs) which are excluded because there is no real risk and 2) foods where there might be a risk in some populations (e.g., seabird eggs, seabirds, possibly other foods linked to aquatic food webs) but where there is unfortunately limited data.

“We estimate the MeHg exposure from seafood consumption for the US population is 11 µg kg body wt⁻¹ a⁻¹”

- Per capita I assume

“We estimate a total of 500,000 points per year of IQ decrements in the US”

- Over what time horizon? Between now and 2050?

“It is not surprising that our results agree better with blood Hg than hair Hg, as the former reflects a shorter-term exposure while the latter subjects to more intrinsic (such as genetics) factors (Basu et al., 2018; Eagles-smith et al., 2018).”

- It is possible that blood Hg provided a better fit than hair Hg because you aggregated exposures on a population basis, averaging out the error introduced by the time lag between individual MeHg exposure and individual biomarker analysis. I think on an individual basis, hair Hg can provide a much better characterization of long-term averages, especially for people who don't eat fish most days. (Even if there is greater uncertainty in the pharmacokinetic parameters for reasons like genetics.) Your assessment above suggests that blood Hg is a better measure than hair Hg in all cases, which I don't think is true. I would suggest rephrasing.

“The economic valuation of these two health endpoints relies on the projection of the global economy, which we adopt the middle-of-the-road pathway projected by the Shared Socioeconomic Pathways

(SSP2) in the 21st century (<https://tntcat.iiasa.ac.at/SspDb>)."

- This is somehow ungrammatical. I think it should be two sentences. "The economic valuation of these two health endpoints relies on the projection of the global economy. We adopt the middle-of-the-road..."

- Review citation style – not sure if in-text hyperlinks are OK.

"We find that the dose-response functions between MeHg intake and health effects have the largest contribution to the uncertainty, ranging from nearly \$0 to \$48 trillion. This reflects the large variability in the coefficients for IQ decrement and heart attack risk per unit hair Hg increase (Axelrad et al., 2007; Rice et al., 2010; Salonen et al., 1995)"

- The inclusion of 0 makes me wonder if you are considering ranges for plausible parameter values that are too wide (i.e., it is beyond all doubt that low levels of MeHg exposures have impacts on neurodevelopment and this manifests in IQ point losses therefore the economic impact should be much bigger than 0).

- Note that Axelrad et al. 2007 did not consider possible confounding with n-3 fatty acids and this has the potential to bias the dose-response relationship downward. (This was pointed out in the Rice et al. 2010 paper you cite.)

- The Salonen 1995 study for cardiovascular effects is by now very out of date. I was surprised to see that you did not cite the 2007 Virtanen study for cardiovascular effects:

<https://www.sciencedirect.com/science/article/pii/S0955286306001008?via=ihub>

- It is possible that pooling estimates for dose-response functions across many decades of papers is introducing a downward bias in your estimates. Many of the older papers are known to underestimate risks.

"We show that the annual global health risk associated with MeHg exposure is \$117 billion (2020 value), contributed by 1.2×10^7 points of IQ loss and 29,000 heart attack deaths."

- \$117 billion per year but are the IQ losses and heart attack deaths also per year? Or is that over some horizon, e.g., by 2050?

"The mean Hg levels vary for ~10 times between the lowest and highest trophic levels, much larger than the impact of water types and whether wild-caught or farm-raised (Figure S7)."

- "large" should be "larger"

"Exceptions are for countries with the least MeHg exposure as found in this study, such as Ethiopia, Tajikistan, and Afghanistan, which are listed as the countries with serious levels of hunger (Global Hunger Index: <https://www.globalhungerindex.org>). In these countries, the MeHg exposure risk of fish consumption is surpassed by its nutrient benefits (Mozaffarian & Rimm, 2006)"

- This seems to say that only in countries with severe hunger and low MeHg exposures do benefits of fish outweigh risks. I strongly disagree with this. In almost all cases, benefits of fish outweigh risks especially considering alternative sources of protein. The relevant risk-risk calculations have been done in a few papers by now: <https://www.ncbi.nlm.nih.gov/pmc/articles/PMC2649230/>
<https://www.ncbi.nlm.nih.gov/pmc/articles/PMC6317887/>

4. Methods

"Due to the large concentration variability, we group the fish/aquatic animals into four trophic level bins: 2-2.5, 2.5-3.5, 3.5-4.5, and 4.5-5, and the geometric mean of MeHg concentrations for each trophic level bin is calculated."

- Why did you group species by trophic bins instead of tracking MeHg for individual species from your species database? I understand for species with no available MeHg data you may use these trophic bins to fill in that missing data, but why did you use trophic bins for species even if you had species-specific data? Wouldn't that introduce more uncertainty than using species-specific data?

- I know you discussed exposure model validation in the results and discussion but it would be good to repeat it or refer to it again here to remind the reader that this approach worked for you. There is so much spatial variability in Hg content even for a given species (never mind for trophic position) that I am honestly surprised your exposure model worked as well as it did.

"We scale the future population exposure of MeHg based on the exposure level at present-day and the model projected environmental Hg levels (Figure 7)."

- The model **for** projected environmental Hg levels?
 “The coefficients β ($0.6 \mu\text{g L}^{-1}$ per $\mu\text{g day}^{-1}$), λ ($0.2 \mu\text{g g}^{-1}$ per $\mu\text{g L}^{-1}$), and γ (0.3 IQ points per $\mu\text{g g}^{-1}$) converse from”
- Converse should be convert
- “...converses hair MeHg concentrations to fatal heart attack risks.”
- Converse should be converts

Reviewer #2 (Remarks to the Author):

Zhang et al. presented in this paper results on an important topic of Global Health Effects of Future Atmospheric Mercury Emissions. This study builds a comprehensive model framework and concludes that there will be a significant increase in global human health cost if emission reduction actions are delayed. I find that the models and input data used in this study are up-to-date. The only major comment that I have is about the uncertainty quantification. Figure 5 shows the range in cumulative health impacts to 2050 from the different scenarios considered in this study. I do not get the method in the calculation of the uncertainty mentioned in the abstract, for the CP scenario, the range is 10-27 trillion dollars. From the data shown in Figure 5, it seems to me that the uncertainty should be much larger if considering different factors, especially dose-response relationship. My understanding is that it is important to consider a comprehensive uncertainty quantification method in the calculations.

Response to comments

Reviewer 1

Comments	Response
The submission by Zhang et al. is an important contribution which will be of interest to a global audience. The article effectively communicates the magnitude of the health and economic burdens created by worldwide mercury emissions. The article is important because it is a rare example of a quantitative model that integrates fundamental processes from emissions to transport to exposure to health impact to economic damages. This submission is likely to be highly influential from a decision/policy support perspective.	We thank the reviewer for the recognition of the importance of this study and the helpful comments/suggestions.
My biggest concern is that the authors sometimes describe fish consumption only in terms of its risk profile from MeHg without considering that, for most people, fish consumption is overall a good thing (even considering MeHg exposures). For example, the authors seem to suggest that only in countries facing hunger crises do the benefits from fish outweigh the risks. I am not sure if this is what the authors meant to say but I strongly disagree with that message. I provide several suggestions below about how this message can be reframed/fine-tuned to capture more of the nuances around benefits from fish in view of risks from MeHg. I believe all of those comments can be addressed with no impact on the quantitative analysis.	We thank the reviewer to bring the nutrient value of fish consumption up. We agree with the reviewer and changed the manuscript accordingly. Please refer to our responses below regarding this issue.
I have also included several other miscellaneous technical comments/questions (again, mostly on the exposure/health side) that could be opportunities for the authors to clarify their methods or possibly adjust their analysis slightly. Overall, I highly encourage you to accept the submission by Zhang et al. subject to the revisions noted below.	Please refer to our responses below.
1. Abstract “the most optimistic scenario (maximum feasible 	The sentence was modified as: “... half of the present-day levels by 2050.” 

reduction, MFR) leads to Hg levels in the freshwater and marine biota half of the present-day levels.”  • By when? By 2050? 	
2. Introduction “Mercury (Hg) is a global pollutant that is associated with impaired neurocognitive deficits in human fetuses and cardiovascular effects in adults (Axelrad et al., 2007; Roman et al., 2011).”  • You mean neurocognitive deficits/impacts or impaired neurodevelopment. An impaired deficit would be a good thing! • I assume you are referring here to impacts of relatively low levels of methylmercury specifically. Higher MeHg exposures can cause neurological impacts even among adults. Other species of Hg (e.g., inhalation of Hg⁰ vapor) also has various impacts, including neurological impacts. • It is important at this stage to know whether you are focusing specifically on MeHg or whether you are talking more broadly about all Hg exposures. 	We modified this sentence as: “Mercury (Hg) is a global pollutant, and its organic form, methylmercury (MeHg) is associated with neurocognitive deficits in human fetuses and cardiovascular effects in adults (Axelrad et al., 2007; Roman et al., 2011).” This also states that we focus on MeHg exposure. We also specified that we are focusing specifically on MeHg exposure in the next sentence: “Human exposure to MeHg is predominantly via the consumption of food (e.g., seafood and rice) (Bellanger et al., 2013; Zhang et al., 2010).”
“Human exposure to Hg is predominantly via the consumption of food (e.g., seafood and rice) that contain methylmercury (MeHg), the most toxic form of Hg (Bellanger et al., 2013).”  • Unless I am mistaken, the Bellanger reference does not say anything about non-MeHg exposures to Hg. Therefore, it can’t be used to support a statement that most Hg exposures are from MeHg. As far as I read, it only says that most MeHg exposures are from seafood. 	The sentence was modified as: “Human exposure to MeHg is predominantly via the consumption of food (e.g., seafood and rice) (Bellanger et al., 2013; Zhang et al., 2010).”
“The annual death from the fatal heart attack that is attributable to MeHg exposure is estimated to be over 10,000 in China and the U.S. (Chen et al., 2019; Giang & Selin, 2016).”  • Can you double check this reference? As far as I read, Giang and Selin do not provide an estimate for the baseline fatalities from MeHg exposure. 	The sentence was modified as: “The annual death from the fatal heart attack that is attributable to MeHg exposure is estimated to be over 10,000 in China (Chen et al., 2019).”
3. Results and discussion	Revised as suggested.

“lifelong earn loss”  • “earn” should be “earnings” 	
“freshwater fish consumption are not negligible in some countries”  • *Non*-negligible? 	Revised as suggested.
“The highest seafood MeHg exposure is found in countries with large seafood consumption, such as the Maldives (33 µg/d),...”  • Is this µg/day per capita? 	We clarified it by adding “per-capita” in this sentence: “The highest per-capita seafood MeHg exposure ...”
“We find that the population's dietary choices are vital factors influencing MeHg exposure and health risk.”  • This paragraph is slightly problematic because it presents a one-dimensional view of fish intake as a vector for MeHg risk. Yes, people who eat the most fish will generally have the highest MeHg exposures, but it does not necessarily follow that they will have the greatest overall risks; depending on the fish they consume, it is likely that, overall, intake of n-3 fatty acids, vitamin D and other nutrients will exert net health protective effects considering relevant alternative sources of protein. (For example, see Calder et al. 2019 where epidemiological modeling demonstrated that even in settings of elevated MeHg, reducing fish consumption almost certainly increases overall risks: https://www.ncbi.nlm.nih.gov/pmc/articles/PMC6317887/). A better framing for this paragraph would be to observe that individuals with the highest fish consumptions and highest MeHg intakes would have the most to gain from reduced Hg emissions and MeHg exposures because they would benefit from greater net benefits of fish intake (risks would go down while benefits would stay the same for a given intake of fish). 	We deleted the phrase “dietary choices” that implies to eat less fish due to MeHg exposure. This sentence was modified as: “We find that the MeHg exposure and health risk are associated with the food intake structures of different countries.” We also rewrote the second paragraph in the “Policy implications” section: “Food intake structure is an important factor for MeHg exposure and risk. Globally, seafood consumption contributes 56% to the total MeHg exposure, with freshwater fish and rice contributing 34% and 10%, respectively. Coastal and island countries with access to more seafood have the largest seafood consumption and they will have the greatest health benefits if Hg emissions are reduced in the future. Freshwater fish consumption is the highest in Asian countries, where fish is often raised in rice paddies (Halwart & Gupta, 2004). The rice consumption in these countries is also high. Despite the elevated MeHg exposure risk, the overall health effects of fish consumption may be positive if considering the intake of n-3 polyunsaturated fatty acids, vitamins, and other nutrients (Calder et al., 2019). Another important influencing factor is the trophic level of fish/aquatic animals. The mean Hg levels vary for ~10 times between the lowest and highest trophic levels, much larger than the impact of water types and whether wild-caught or farm-raised (Figure S7). Dietary guidance on fish selection but not the total fish

	consumption is the rule of thumb to minimize the overall health risks, especially considering the nutrient effects of fish (Xue et al., 2015; Gindberg and Toal, 2009). For countries with the least MeHg exposure as found in this study, such as Ethiopia, Tajikistan, and Afghanistan, which are listed as the countries with serious levels of hunger (Global Hunger Index: https://www.globalhungerindex.org). In these countries, the MeHg exposure risk of fish consumption is even more outweighed by its nutrient benefits (Mozaffarian & Rimm, 2006). We suggest that the Hg level in rice is the most recalcitrant to emission reduction among the three major food categories, and the global contribution from rice consumption could increase to 23% in 2050 under the MFR scenario, which makes limiting rice consumption may be a more important Hg exposure mitigation strategy then.”
“we exclude the exposure from other food such as pork, beef, and eggs, which have limited MeHg concentration data”  • You should say that you exclude these foods not because they have limited data but because MeHg is negligible compared to fish and seafood. (This is the reason there is limited data – they are not important contributors to overall MeHg exposures.) The exception here could be eggs and meat from seabirds like loons and other birds that eat fish. In some populations (e.g., Indigenous people in North America) these can be important contributors. I agree that for these foods, there might be real exposures but there is limited data. I know there is lots of loon Hg/MeHg data available for northeastern North America (mostly by Dave Evers, e.g., https://link.springer.com/article/10.1023/A:1022593030009) but data for other seabirds has been hard to find in my experience. It may be good to make a distinction between 1) animals with no link to aquatic food webs (e.g., pork, beef, most eggs) which are excluded because	This sentence was modified as: “...we exclude the exposure from other food such as pork, beef, and eggs, which have negligible contribution to the total exposure (except for the eggs and meat from fish-eating seabirds that are consumed by some indigenous populations, e.g. Evers et al., 2003, but there is limited data and may not be important for general populations).”

there is no real risk and 2) foods where there might be a risk in some populations (e.g., seabird eggs, seabirds, possibly other foods linked to aquatic food webs) but where there is unfortunately limited data.	
“We estimate the MeHg exposure from seafood consumption for the US population is 11 µg kg body wt-1 a-1”  • Per capita I assume 	Yes, it is. We clarified it by modifying this sentence as: “We estimate the per-capita MeHg exposure ...”
“We estimate a total of 500,000 points per year of IQ decrements in the US”  • Over what time horizon? Between now and 2050? 	We clarified it by modifying this sentence as: “We estimate a total of 500,000 points per year of IQ decrements in the US at present-day, ...”
“It is not surprising that our results agree better with blood Hg than hair Hg, as the former reflects a shorter-term exposure while the latter subjects to more intrinsic (such as genetics) factors (Basu et al., 2018; Eagles-smith et al., 2018).”  • It is possible that blood Hg provided a better fit than hair Hg because you aggregated exposures on a population basis, averaging out the error introduced by the time lag between individual MeHg exposure and individual biomarker analysis. I think on an individual basis, hair Hg can provide a much better characterization of long-term averages, especially for people who don’t eat fish most days. (Even if there is greater uncertainty in the pharmacokinetic parameters for reasons like genetics.) Your assessment above suggests that blood Hg is a better measure than hair Hg in all cases, which I don’t think is true. I would suggest rephrasing. 	We agree with the reviewer and we rephrased this sentence as: “In addition to the MeHg exposure, the biomarker level subjects to the variability of pharmacokinetic and intrinsic (such as genetics) factors (Basu et al., 2018; Eagles-smith et al., 2018).”
“The economic valuation of these two health endpoints relies on the projection of the global economy, which we adopt the middle-of-the-road pathway projected by the Shared Socioeconomic Pathways (SSP2) in the 21st century (https://tntcat.iiasa.ac.at/SspDb).”  • This is somehow ungrammatical. I think it should be two sentences. “The economic valuation of these two health endpoints relies on the projection of the global economy. We adopt the middle-of-the-road...” • Review citation style – not sure if in-text 	We split the sentence in two as suggested. We also checked the citation style and Nature Communications allows in-text hyperlinks.

hyperlinks are OK.	
“We find that the dose-response functions between MeHg intake and health effects have the largest contribution to the uncertainty, ranging from nearly \$0 to \$48 trillion. This reflects the large variability in the coefficients for IQ decrement and heart attack risk per unit hair Hg increase (Axelrad et al., 2007; Rice et al., 2010; Salonen et al., 1995)”  • The inclusion of 0 makes me wonder if you are considering ranges for plausible parameter values that are too wide (i.e., it is beyond all doubt that low levels of MeHg exposures have impacts on neurodevelopment and this manifests in IQ point losses therefore the economic impact should be much bigger than 0). • Note that Axelrad et al. 2007 did not consider possible confounding with n-3 fatty acids and this has the potential to bias the dose-response relationship downward. (This was pointed out in the Rice et al. 2010 paper you cite.) • The Salonen 1995 study for cardiovascular effects is by now very out of date. I was surprised to see that you did not cite the 2007 Virtanen study for cardiovascular effects: https://www.sciencedirect.com/science/article/pii/S0955286306001008?via=ihub • It is possible that pooling estimates for dose-response functions across many decades of papers is introducing a downward bias in your estimates. Many of the older papers are known to underestimate risks. 	Thanks for the reviewer to bring it up. We used a narrower distribution for the coefficients of heart attack risk per unit hair Hg increase as suggested by more up-to-date references: “We find that the dose-response functions between MeHg intake and health effects have the largest contribution to the uncertainty, ranging from nearly \$7.8 to \$47 trillion. This reflects the large variability in the coefficients for IQ decrement and heart attack risk per unit hair Hg increase (Virtanen et al., 2007; Giang and Selin, 2016).” We still find a relatively wide range for this parameter, consistent with previous studies (e.g. Giang and Selin, 2016). So, our wordings can keep unchanged. We also updated Figure 5 with this new result:  Figure 5. Range in cumulative health impacts to 2050 for the CP, MFR, NP-Delayed, A1B, and A2 scenarios. Bars indicate the sensitivity of cumulative health effects to high and low case assumptions for uncertain parameters (as 95% confidence intervals): food consumption, economic valuation, dose-response parameterization, and food Hg concentrations. The black lines are our best estimates.
“We show that the annual global health risk associated with MeHg exposure is \$117 billion (2020 value), contributed by 1.2×10^7 points of IQ loss and 29,000 heart attack deaths.”	Sorry for the confusion. We clarified this sentence as: “We show that the annual global health risk associated with MeHg exposure at present-day is

 • \$117 billion per year but are the IQ losses and heart attack deaths also per year? Or is that over some horizon, e.g., by 2050? 	\$117 billion (2020 value), contributed by 1.2×10^7 points of IQ loss and 29,000 heart attack deaths per year."
“The mean Hg levels vary for ~10 times between the lowest and highest trophic levels, much larger than the impact of water types and whether wild-caught or farm-raised (Figure S7).”  • “large” should be “larger” 	Revised as suggested.
“Exceptions are for countries with the least MeHg exposure as found in this study, such as Ethiopia, Tajikistan, and Afghanistan, which are listed as the countries with serious levels of hunger (Global Hunger Index: https://www.globalhungerindex.org). In these countries, the MeHg exposure risk of fish consumption is surpassed by its nutrient benefits (Mozaffarian & Rimm, 2006)”  • This seems to say that only in countries with severe hunger and low MeHg exposures do benefits of fish outweigh risks. I strongly disagree with this. In almost all cases, benefits of fish outweigh risks especially considering alternative sources of protein. The relevant risk-risk calculations have been done in a few papers by now: https://www.ncbi.nlm.nih.gov/pmc/articles/PMC2649230/ https://www.ncbi.nlm.nih.gov/pmc/articles/PMC6317887/ 	Please refer to our response above.
4. Methods “Due to the large concentration variability, we group the fish/aquatic animals into four trophic level bins: 2-2.5, 2.5-3.5, 3.5-4.5, and 4.5-5, and the geometric mean of MeHg concentrations for each trophic level bin is calculated.”  • Why did you group species by trophic bins instead of tracking MeHg for individual species from your species database? I understand for species with no available MeHg data you may use these trophic bins to fill in that missing data, but why did you use trophic bins for species even if you had species-specific data? Wouldn't that introduce more uncertainty than using species-specific data? 	Thanks for the reviewer to bring it up. We indeed have collected a lot of MeHg concentrations data for individual fish/aquatic animal species. However, the database for fish/aquatic animal consumption (i.e the UN FAO database) only reports a total consumption of fish/aquatic animals. We tried to collect consumption data for individual species, but they are only available for a small subset of the countries (and for most of the time they are not comparable with each other). So, we compromised to group fish MeHg concentrations to trophic level bins, and separate the total fish consumption to trophic level bins based on marine trophic index data.

• I know you discussed exposure model validation in the results and discussion but it would be good to repeat it or refer to it again here to remind the reader that this approach worked for you. There is so much spatial variability in Hg content even for a given species (never mind for trophic position) that I am honestly surprised your exposure model worked as well as it did.	We clarified this point by modifying this sentence as: “Due to the large concentration variability and the lack of fish/aquatic animals consumption data for individual species, we group the fish/aquatic animals into four trophic level bins ...” We also modified the last sentence of the paragraph above as: “The per-capita consumption of different food categories (including rice, total fish and aquatic animals) for each country is taken from the database of the Food and Agriculture Organization of the United Nations (UN FAO, http://www.fao.org).” We are also thrilled by the good agreement between our model results and human biomarker data. This suggests that our approach (even though highly simplified) indeed capture important influencing factors for the large-scale variability of MeHg exposure in the general population. We added a sentence at the end of this paragraph: “The agreement with human biomarker data suggests that our simplified exposure model works reasonably well.”
“We scale the future population exposure of MeHg based on the exposure level at present-day and the model projected environmental Hg levels (Figure 7).” • The model *for* projected environmental Hg levels?	We revised this sentence as: “We scale the future population exposure of MeHg based on the exposure level at present-day and the model-projected environmental Hg levels (Figure 7).”
“The coefficients β ($0.6 \mu\text{g L}^{-1}$ per $\mu\text{g day}^{-1}$), λ ($0.2 \mu\text{g g}^{-1}$ per $\mu\text{g L}^{-1}$), and γ ($0.3 IQ$ points per $\mu\text{g g}^{-1}$) converge from” • Converge should be convert	Revised as suggested.
“...converses hair MeHg concentrations to fatal heart attack risks.” • Converses should be converts	Revised as suggested.

Reviewer 2

Comments	Response
Zhang et al. presented in this paper results on an important topic of Global Health Effects of Future Atmospheric Mercury Emissions. This study builds a comprehensive model framework and concludes that there will be a significant increase in global human health cost if emission reduction actions are delayed. I find that the models and input data used in this study are up-to-date.	We thank the reviewer for the recognition of the importance of this study and the helpful suggestions.
The only major comment that I have is about the uncertainty quantification. Figure 5 shows the range in cumulative health impacts to 2050 from the different scenarios considered in this study. I do not get the method in the calculation of the uncertainty mentioned in the abstract, for the CP scenario, the range is 10-27 trillion dollars. From the data shown in Figure 5, it seems to me that the uncertainty should be much larger if considering different factors, especially dose-response relationship. My understanding is that it is important to consider a comprehensive uncertainty quantification method in the calculations.	We apologize for this mistake. The range of \$10-27 trillion considers only the uncertainty from “food consumption” (i.e. blue bar for the CP scenario). We corrected this error and calculated the comprehensive uncertainty in the revised manuscript: We added these sentences at the end of Method section: “The overall uncertainty is estimated by a Monte Carlo approach. The health risk calculation is repeated for 1000 times with randomly sampled parameters for these four factors following Chen et al. (2019). The 2.5% and 97.5% percentiles of the calculated risk are taken as the overall uncertainty range (i.e. 95% confidence interval).” We added a sentence in line 296: “By taking a Monte Carlo approach (see Methods), we also calculate the overall uncertainty range as \$4.7-54 trillion.” The uncertainty range was also updated in the abstract: “Our results show that the accumulated health effects associated with Hg exposure during 2010-2050 are \$19 (95% confidence interval: 4.7-54) trillion ...” The first paragraph of the “Uncertainty” section and Figure 5 were also updated accordingly. Please refer

	to the tracked version of the revised manuscript for more details.
--	--

REVIEWERS' COMMENTS

Reviewer #1 (Remarks to the Author):

The authors have adequately addressed all my comments. They are to be commended for their great work!

Reviewer #2 (Remarks to the Author):

The authors have well addressed my comments

Response to reviewers

Reviewer 1:

C: The authors have adequately addressed all my comments. They are to be commended for their great work!

R: We thank again the reviewer for the helpful comments and suggestions.

Reviewer 2:

C: The authors have well addressed my comments.

R: We thank again the reviewer for the helpful comments and suggestions.